# Obesity Modulates the Gut Microbiome in Triple-Negative Breast Cancer

**DOI:** 10.3390/nu13103656

**Published:** 2021-10-19

**Authors:** Fokhrul Hossain, Samarpan Majumder, Justin David, Bruce A. Bunnell, Lucio Miele

**Affiliations:** 1School of Medicine, Louisiana State University Health Sciences Center, New Orleans, LA 70112, USA; fhossa@lsuhsc.edu (F.H.); smaju1@lsuhsc.edu (S.M.); jdav45@lsuhsc.edu (J.D.); 2Department of Microbiology, Immunology and Genetics, University of North Texas Health Sciences Center, Fort Worth, TX 76107, USA; Bruce.Bunnell@unthsc.edu

**Keywords:** triple-negative breast cancer (TNBC), commensal microbiota, 16S rRNA sequencing, metagenomic analyses

## Abstract

Triple-negative breast cancer (TNBC) is an aggressive, molecularly heterogeneous subtype of breast cancer. Obesity is associated with increased incidence and worse prognosis in TNBC through various potential mechanisms. Recent evidence suggests that the gut microbiome plays a central role in the progression of cancer, and that imbalances or dysbiosis in the population of commensal microbiota can lead to inflammation and contribute to tumor progression. Obesity is characterized by low-grade inflammation, and gut dysbiosis is associated with obesity, chronic inflammation, and failure of cancer immunotherapy. However, the debate on what constitutes a “healthy” gut microbiome is ongoing, and the connection among the gut microbiome, obesity, and TNBC has not yet been addressed. This study aims to characterize the role of obesity in modulating the gut microbiome in a syngeneic mouse model of TNBC. 16S rRNA sequencing and metagenomic analyses were performed to analyze and annotate genus and taxonomic profiles. Our results suggest that obesity decreases alpha diversity in the gut microbiome. Metagenomic analysis revealed that obesity was the only significant factor explaining the similarity of the bacterial communities according to their taxonomic profiles. In contrast to the analysis of taxonomic profiles, the analysis of variation of functional profiles suggested that obesity status, tumor presence, and the obesity–tumor interaction were significant in explaining the variation of profiles, with obesity having the strongest correlation. The presence of tumor modified the profiles to a greater extent in obese than in lean animals. Further research is warranted to understand the impact of the gut microbiome on TNBC progression and immunotherapy.

## 1. Introduction

The human large intestine harbors 10^13^–10^14^ bacterial microorganisms, which compose the gut microbiome [1,2]. The gut microbiome plays essential roles in maintaining cellular metabolism and physiology in conjunction with human cells [1,3]. The composition of the gut microbiome is dependent on natural selection [2]. It is dynamic, beginning after birth and continuously changing in response to diet [3] and environmental factors, such as medication use [4]. Multiple microbiomes exist in the human body, and their anatomical location and function determine the degree of bacterial diversity. For example, the microbiome of a healthy gut has a high degree of diversity, while a healthy vaginal microbiome has low diversity [5].

The gut microbiome is also important in the regulation of the host immune system. If the typical microbial balance is disrupted, a state of dysbiosis occurs, which can lead to systemic inflammation [6] through a variety of mechanisms, including the activation of Toll-like receptors in the innate immune system [7,8]. Inflammation is involved in the progression of cancers, including breast and colon cancer [4,6,9]. There is evidence that the composition of the microbiome affects cancer immunity and the response to cancer immunotherapy, as reviewed in [10]. The absence of specific microbes can be associated with the alteration of gut lymphoid tissue integrity [11]. Gut microorganisms have been shown to have potential cancer-preventive properties as well; *Bacteroides fragilis*, for instance, produces a polysaccharide that has been documented to correct host T-cell deficiencies in germ-free mice [12]. These findings suggest a connection between the gut microbiome and cancer, but a direct link [4,6] and the constitution of a “healthy” gut microbiome [11] have not been clearly defined.

Breast cancer is the most common cancer of women worldwide. Gut microbiome alterations play a role in breast cancer [6,13], but the mechanisms remain unclear. Estrogen receptor-positive breast cancer is more likely associated with a hyperactive estrobolome, the genes in commensal bacteria that code for estrogen metabolizing proteins. A high level of deconjugation causes increased intestinal absorption of free estrogens, which increases the risk for breast cancer [14]. Women with breast cancer have different microbiota than women without cancer [15], so defining the gut microbiome composition and stratifying patients based on its composition could be beneficial for diagnostic and therapeutic purposes [7,15]. Incorporating gut microbiome analyses into precision medicine protocols along with host genomics and environmental exposures is rapidly gaining interest [10].

Obesity is one of the most prevalent comorbidities worldwide, especially in the United States. Numerous causes and lifestyle factors, including physical activity, contribute to obesity, but an imbalance of caloric intake is a primary factor [16]. Dietary factors are thought to cause about 4% of cancers [17,18]. Obesity is an established risk factor for at least 12 cancer types and a likely risk factor for several others [19]. Being overweight is also related to 15–20% of cancer deaths [20]. Obesity is highly correlated with the incidence of postmenopausal breast cancer. Twenty percent of breast cancer cases in postmenopausal women and fifty percent of breast cancer deaths are attributable to obesity [21]. A meta-analysis concluded that a greater waist-to-hip ratio (WHR) increased the risk of breast cancer, while a reduction in WHR decreased it [22]. High body weight was also associated with more progesterone receptor-positive breast cancer [23,24] and estrogen receptor-positive breast cancer [23]. The metabolic and endocrine effects of obesity lead to increased production of steroid hormones and peptide hormones, such as leptin and VEGF, which have profound effects on breast cancer biology [20]. The mechanisms whereby obesity contributes to breast cancer risk and outcomes are diverse and likely different for different subtypes of breast cancer.

Triple-negative breast cancer (TNBC) is a definition encompassing all breast cancers with no/low immunohistochemically detectable expression of estrogen receptor α and progesterone receptor, and a lack of genomic amplification of ERBB2/HER2 [25,26]. TNBCs are clinically aggressive and typically affect premenopausal women, especially African American women [27,28,29]. Currently, chemotherapy and immunotherapy in a subset of tumors expressing PD-L1 are the main pharmacological options for the treatment of TNBC [30,31]. The gut microbiome composition has been reported to affect the efficacy of chemotherapy [32,33,34]. Obesity is associated with increased incidence of TNBC and poor prognosis for TNBC patients [35]. Multiple mechanisms are likely to contribute to these effects, including increased systemic inflammation, reactive oxygen species, leptin, hyperinsulinemia and altered metabolism [36]. A diet including high amounts of animal products, such as animal fat, eggs, and meat, was positively correlated with TNBC, while a plant-based diet of vegetables, vegetable fat, and nuts is negatively correlated with TNBC [37]. Unlike other types of breast cancer, obesity is associated with an increased risk of TNBC in premenopausal women [38]. The mechanisms of obesity-modulated TNBC progression remain poorly understood.

Since dysbiosis in the gut microbiome and obesity are correlated with each other, it is important to understand the potential role played by the gut microbiome and in the crosstalk between obesity and TNBC in tumor-bearing animal models and cancer patients. We studied the gut microbiome in lean and obese tumor-free and tumor-bearing immune-competent mice with a transplantable syngeneic TNBC model, using a “Western” diet to induce obesity. Western diet-induced obesity dramatically decreased microbiome alpha diversity in both tumor-free and tumor-bearing mice, with a significant decrease in *Alistipes*. The gut microbiome of obese tumor-bearing mice was relatively enriched in *Firmicutes* such as *Clostridia* and *Mogibacteriaceae* compared to lean tumor-bearing mice. Obesity significantly explained differences in taxonomic profiles. Obesity, tumor presence, and obesity–tumor interaction significantly explained differences in the functional profiles of gut microbiomes.

## 2. Methods and Materials

### 2.1. Diet-Induced Obesity and Experimental Design

Mice were fed ENVIGO TD.88137, an adjusted calorie diet that mimics the “Western diet”. It is not a high-fat diet, but a balance of fat (42.0% of kcal, of which >60% are saturated fatty acids) and carbohydrates (42.7% of kcal, including 34% sucrose by weight), with protein constituting 15.2% of kcal (ENVIGO, Indianapolis, IN, USA). This diet accelerates and enhances atherosclerosis and plaque formation, leading to obesity. Due to the balance of fats and carbohydrates, this diet was chosen because it best simulates the human “Western diet”. FVB female mice were kept on a regular control diet (crude protein: 19%; fat: 9%; carbohydrate: 44.9% (2019S, ENVIGO)) vs. “Western Diet” for four months. Mouse body weight was monitored throughout this time. Then, 1 million syngeneic C0321 mouse TNBC cells [39] were injected into the mammary fat pads of mice with Matrigel (1:1 ratio), and tumor growth was monitored for 3 weeks. After three weeks, tumors were harvested for downstream processing, and fecal samples from the large intestine (colon) were collected for microbiome analysis. 16s rRNA sequencing and metagenomics analyses were performed at Microbiome Insights (Vancouver, BC, Canada).

### 2.2. 16s rRNA Sequencing

Bacterial 16s rRNA genes (V4 region) were sequenced on an Illumina MiSeq. Raw Fastq files were quality-filtered and clustered into 97% similarity operational taxonomic units (OTUs) using the mothur software package by Microbiome Insights. High-quality reads were classified using the Greengenes reference database. We obtained a consensus taxonomy for each OTU. We then aggregated OTU abundances into taxonomies and plotted the relative abundances of the most abundant ones. OTU abundances were converted into pairwise dissimilarities (Bray–Curtis index). Multidimensional scaling (MDS) was used to visualize microbiome similarities in ordination plots. Permutational analysis of variance (PERMANOVA) was used to test for the significance of microbiome differences. Negative binomial tests (DESEq2 R package) were performed for differential abundance analysis. Alpha diversity was calculated using Shannon’s diversity index.

### 2.3. Metagenomics

#### 2.3.1. DNA Extraction and Library Preparation

DNA was extracted using the Qiagen’s (Germany) MagAttract PowerSoil DNA KF kit (Formerly MO Bio PowerSoil DNA Kit) using a KingFisher robot at Microbiome Insights. DNA quality was evaluated using gel electrophoresis and then quantified using a Qubit 3.0 fluorometer (Thermo-Fischer, Waltham, MA, USA). DNA Libraries were generated using an Illumina Nextera library preparation kit following the standard protocol (Illumina, San Diego, CA, USA).

#### 2.3.2. Sequence Technology and Processing

Sequencing was performed using an Illumina NextSeq. Around 25.87 Gbases were generated using 2 × 150 paired-end reads. Each sample yielded a median of 1.14 Gbases, which was very close to our intended target of 1 Gbase per fecal sample. After the sequencing, reads were arranged based on the barcodes. Initial quality was evaluated using FastQC v0.11.5. There were three steps in the data processing: (a) paired-end read joining, (b) removal of contaminants, and (c) trimming. The paired-end reads were joined using FLASH v1.2.11 [40]. Reads were then sequentially compared to the mouse and human genomes (Genome Reference Consortium Mouse strain FVB_NJ, Genome Reference Consortium Mouse Build 38 patch release 6, and Genome Reference Consortium Human Reference 37), and sequences mapped to them were removed. Finally, sequences were trimmed according to their quality values using Trimmomatic v0.36 [41]. The paired read joining reduced the library size on average by 29.82%. Furthermore, 39.05% of the stitched reads mapped to the mouse and human genomes were removed. Read trimming using quality filters removed 7.47% of the screened reads. At the end of quality control, the median number of quality-filtered reads per sample was 3814,405.

### 2.4. Taxonomic and Functional Analyses

Taxonomic composition was determined using Metaphlan2 [42]. We used an ordination approach to obtain a graphical representation of similarity among samples. The similarity between any two samples, based on their microbiome compositions, was calculated using Bray–Curtis dissimilarities, which considers both presence/absence and abundance of the species. The distances were then evaluated and represented graphically using non-metric multidimensional scaling (NMDS) ordination. Permutational multivariate analysis of variance (PERMANOVA) determined the significance of differences between treatments. The R2 represents, when significant, the proportion of the variability that is explained by those factors. Using an ordination approach, the taxonomic profiles were created as a graphical representation of similarity among samples. Ordination plots were arranged by color according to tumor presence and by shape according to the obesity of the mice.

Functional profiles were summarized into pathways using the Metacyc pathway definition. The difference in pathway richness (the number of unique pathways) was calculated using the Scheirer–Ray–Hare test (nonparametric test for a two-way factorial design). Functional profiles include the gene family profiles generated using the Uniref database. These genes are then integrated into different gene groups: Metacyc pathways, Metacyc reactions, KEGG orthogroups (KOs), Pfam domains, level-4 enzyme commission (EC) categories, EggNOG (which includes COGs), Gene Ontology (GO), and Informative GOs.

Permutational multivariate analysis of variance using distance matrices (PERMANOVA, using the *adonis* R function) determined the significance of differences between treatments. The R2 represented, when significant, the proportion of the variability that was explained by those factors. Residuals represented the unexplained variation. Differential abundance testing was performed using a Scheirer–Ray–Hare test (non-parametric ANOVA for two factors).

## 3. Results

We modeled TNBC in a lean versus obese background by using a Western diet-induced obese mouse model. This diet was chosen because its composition is similar to the standard American diet [43]. Female FVB mice were fed a regular/control vs. Western diet for four months to induce obesity. Lean and obese mice were then injected with syngeneic mouse TNBC cells, and the tumors were allowed to develop for three weeks, at which point, the tumors were collected for downstream processing, and fecal samples from the colon were collected for 16s rRNA sequencing and metagenomics analysis. Western diet-induced mouse body weight gain was expected (Appendix A). Weight differences between Western diet-fed mice and control mice remained virtually identical in tumor-free and tumor-bearing animals. Importantly, tumors did not induce weight loss (Appendix A). Tumor volumes beginning 10 days after injection and throughout the experiment were significantly higher in obese animals, consistent with an association between obesity and TNBC progression (Appendix A).

### 3.1. Bacterial 16S rRNA Sequencing

Bacterial 16S rRNA genes (V4 region) were sequenced on an Illumina MiSeq and analyzed by Microbiome Insights as described in the Methods section. We had to drop one sample (number 16) due to quality control failure during sample processing at Microbiome Insight. First, we analyzed the community composition of the relative abundances of the most abundant taxa (Figure 1A).

Abundances of bacterial genera varied depending on obesity status and tumor presence. Tumor-free obese mice had a significant loss of diversity and appeared to harbor a relatively high percentage of *Akkermansia* compared to all other groups. This was most likely not due to an absolute increase in *Akkermansia* but to a loss of diversity among other taxa. Obese mice with or without tumor had decreased *Lachnospiraceae* compared to lean mice. Tumor-bearing obese mice had increased proportions of *Clostridiales* such as *Clostridium* and *Mogibacteriaceae* compared to all other groups. Interestingly, the community composition of the most abundant abundant taxa was altered in obese tumor-bearing animals compared to obese tumor-free animals. We identified 12 differentially abundant OTUs according to sorted q-value, namely, *Subdoligranulum*_Otu0044, *Lachnospiracea*e_unclassified_Otu0022, *Coprococcus*_Otu0091, *Clostridium*_Otu0061, *Lachnospiraceae* _unclassified _Otu0081, *Oscillospira*_Otu0070, L *Lachnospiraceae*_unclassified_Otu0028, *Clostridium*_Otu0010, *Ruminococcaceae*_unclassified_Otu0041, *Lactobacillus*_Otu0053, Bacteria_unclassified_Otu0064 and *Lachnospiraceae*_unclassified_Otu0056 (Figure 1B).

Next, the similarity of community compositions between samples was analyzed by non-metric multidimensional scaling (NMDS). The NMDS ordination plot displays the similarity of community compositions among samples (Figure 2A). Microbial communities from each group of samples clustered together, suggesting their similarity. We found four independent clusters in our samples, corresponding to the four treatment groups. The ordination plot showed that microbiome compositions in lean mice with and without tumors were the most similar, while those in obese mice with and without tumors were the most dissimilar. In other words, the presence of tumor modified the intestinal microbiome in obese animals, even though no significant weight loss was observed in tumor-bearing animals.

Finally, the alpha diversity of microbiomes was analyzed as a function of Shannon’s index. We found that obesity decreased alpha diversity in both tumor- and non-tumor-bearing mice samples (Figure 2B, Appendix A). Shannon’s index indicates that lean mice have higher gut microbiome diversity compared to obese mice. Our results suggested that obesity decreased average microbial species diversity in the gut microbiome. The host immune system depends on a healthy gut microbiome that relies on rich and diverse microbial species.

### 3.2. Metagenomic Profiles

Metagenomics analyses were performed, as described in the Methods section. One tumor-bearing obese mouse sample had to be removed from the study due to quality control failure during metagenomic analysis. The taxonomic composition was determined using Metaphlan2. The microbiome community was dominated by bacteria, which accounted on average for 99.885% of the entire community (Appendix A). Among bacteria, the samples were dominated by *Verrucomicrobia*, *Firmicutes*, and *Bacteroidetes* (Appendix A). At the phyla level, obese samples were dominated by *Verrucomicrobia*, with relative loss of *Bacterioidetes* (Appendix A). At the genus level, obese samples (both tumor-bearing and non-tumor-bearing animals) were dominated by *Akkermansia*, with nearly complete loss of *Alistipes* and *Lactobacillus*. (Figure 3).

### 3.3. Community Composition: Visualizing Similarity among Microbiomes

We used an ordination approach to obtain a graphical representation of similarity among samples. The similarity between any two samples was calculated using Bray–Curtis dissimilarities, as described in the Methods section. The distances were then evaluated and represented graphically using non-metric multidimensional scaling (NMDS) ordination (Figure 4). We found separated clusters for lean and obese samples.

However, clusters between tumor-bearing vs. non-tumor bearing were not very distinct in both lean and obese groups. Then, permutational multivariate analysis of variance using distance matrices (PERMANOVA) was used to analyze the variation of taxonomic groups. Obesity was the only significant factor (*p*-value 0.0002), explaining the similarity of bacterial communities according to their taxonomic profiles (Table 1). The tumor–obesity interaction did not reach statistical significance (*p* = 0.0879), but may warrant further investigation with larger sample sizes. We then analyzed taxa that changed significantly according to obesity and/or tumor presence. We determined the differential abundance of taxonomic groups using the Scheirer–Ray–Hare test. Table 2 contains data for species that differed in abundance according to obesity, tumor, or obesity–tumor interaction. Taxa that were changed significantly in association with obesity are listed in Table 2 (first column). *Alistipes*, *Ruminococcus torques*, *Dorea*, *Eubacterium plexicaudatum*, *Lactobacillus johnsonii*, *Lactococcus lactis*, *Oscillibacter*, *Subdoligranulum*, and *Burkholderiales* changed significantly according to obesity. *Akkermansia muciniphila* was the only significant taxa associated with tumor variability. However, when we combined obesity and tumor, the significance of most taxa was lost, except for *Parasutterella excrementihominis*. Adjusted *p* values for several taxa were significant for obesity (Table 2, column 4) but not for tumor or obesity–tumor interaction (Table 2, columns 5 and 6).

### 3.4. Functional Profiles of Bacteria

Functional diversities were summarized into pathways using Metacyc pathway definitions. The difference in pathway richness (number of unique pathways) was calculated using the Scheirer–Ray–Hare test (nonparametric test for a two-way factorial design). We found that obesity had a significant effect on pathway richness (*p*-value: 0.0011), while tumor and obesity–tumor interaction were not statistically significant (*p*-value: 0.622) (Figure 5 and Appendix A). We then analyzed the similarities of functional profiles among samples using an ordination approach, as described earlier. The distances were then evaluated and represented graphically using NMDS ordination (Figure 6).

Permutational multivariate analysis of variance using distance matrices (PERMANOVA) determined the significance of differences between arms. The analysis of variation of functional profiles suggested that the obesity status, tumor presence, and the obesity–tumor interaction were significant in explaining the variation of functional profiles (Table 3). Obesity was the most substantial explanatory factor, accounting for 57.6% of the variation.

Differential abundance testing of functional groups was determined by using a Scheirer–Ray–Hare test (nonparametric ANOVA for two factors). Table 4 shows the top pathways that differed in abundance according to obesity, tumor, or obesity–tumor interaction. Obesity was associated with significant variation in numerous pathways. The most important pathways with significantly different abundance according to obesity were N10-formyl-tetrahydrofolate biosynthesis, homolactic fermentation, arginine biosynthesis I (via L-ornithine), arginine biosynthesis II (acetyl cycle), chorismate biosynthesis I, and others (Table 4). The presence of tumor was significantly associated with glycolysis III (anaerobic glycolysis), and obesity–tumor interaction was significantly associated with glycolysis III, fucose degradation, and the super-pathway of beta D-glucuronide and D-glucuronate degradation (Table 4). Pathway richness analysis of the four groups was also performed.

Figure 5 represents the differences in functional richness or number of unique pathways among the four groups. In our analysis, obesity was the only determinant that affected pathway richness. Figure 6 represents the similarity of functional profiles among the four groups, which is a measure of the convergence of common pathways among the four groups of mice. Once again, obese mice with or without tumors tended to cluster together, and clearly segregated from lean animals.

## 4. Discussion

We examined the gut microbiome composition in lean or obese mice with or without syngeneic TNBC tumors to model tumor progression in lean versus obese patients. Obesity was induced with a “Western” diet, commonly used in atherogenesis and obesity studies. Tumor growth was accelerated in obese mice, suggesting that obesity promoted tumor progression. This is consistent with studies showing that obesity increases the risk of TNBC in women [36]. Possible mechanisms include hypercholesterolemia and hyperinsulinemia [44,45,46], as well as systemic inflammation, which can contribute to the risk of multiple cancers, including breast cancer [4,9].

The analysis of 16s rRNA sequences has revealed the predominant bacterial phyla in the human gut microbiome, including *Firmicutes*, *Bacteroidetes*, *Actinobacteria, Fusobacteria*, *Proteobacteria*, and *Verrucomicrobia* [2,47,48]. * Firmicutes* and *Bacteroidetes* constitute over 90% of the human gut microbiome [2,48]. In addition to bacteria, other microbes, including archaea, eukaryotes, and viruses, are also present in the gut microbiome [49]. Our 16S rRNA sequence analysis revealed variation in microbiomes between tumor-free or tumor-bearing lean and obese mice. Alpha diversity analysis showed that lean mice had higher microbiome diversity than obese mice. This loss of alpha diversity with obesity is consistent with the literature [50]. Functional analysis showed a large number of microbial metabolic pathways significantly altered by obesity. Interestingly, a few pathways (anaerobic glycolysis, fucose degradation, glycuronide/glycuronic acid degradation, peptidoglycan biosynthesis, and CDP-diacylglycerol biosynthesis I) were selectively altered in tumor-bearing animals. The gut microbiome can contribute to the onset of obesity through a variety of mechanisms. The gut microbiome is important to host metabolism and energy storage and can lead to an increase in adiposity and insulin resistance [51]. Microbiota can directly increase the absorption of monosaccharides [51]. Metagenomics analyses show that the microbiome of obese mice has increased energy-harvesting capacity [52]. This is likely to reflect the increased availability of readily metabolized nutrients in obesogenic diets. Intestinal dysbiosis promoted by obesogenic diets can promote obesity through multiple mechanisms [53]. These include, among others, systemic inflammation promoted by endotoxin through Toll-like receptor (TLR) binding, increased insulin secretion, and insulin resistance [53]. Another mechanism for the interplay between the microbiome and obesity is altered bile acid metabolism, which in turn modulates the farnesoid X receptor in the liver [54]. The gut flora may also affect neural feeding behavior through vagal stimulation or immune-neuroendocrine mechanisms [55]. Another possible mechanism includes gut microbiota-induced suppression of fasting-induced activating factor (*Fiaf*), a lipoprotein-lipase inhibitor. The suppression of *Fiaf* increases the deposition of triglycerides into adipose tissue [51,56].

There is controversy surrounding the microbiome composition that contributes to obesity [57,58,59]. One of the most common findings is that obesity is associated with an increase in *Firmicutes* and a decrease in *Bacteroides* [58,59,60,61,62]. In contrast, some results show a decrease in *Firmicutes* and no change in *Bacteroides* [63], and others show an increased ratio of *Bacteroides:Firmicutes* [60]. Another study found that a certain enterotype consisting of a high proportion of *Bacteroides* increases systemic inflammation and obesity progression [64]. In addition, species within genera can have differing effects. In the genus *Lactobacillus*, *L. reuteri* are associated with increased risk of obesity, while *L. casi* and *L. plantarum* are associated with decreased risk for obesity [57]. These varying studies show that defining a microbiome composition that promotes obesity is difficult, as confounding variables are present, including host genotype and diet [57]. Our metagenomic analysis at the phyla level showed a dramatic decrease in *Bacteroides* in obese mice, irrespective of tumor status. This is consistent with a decreased *Bacteroides/Firmicutes* ratio. Obesity was strongly associated with a decrease in *Bacteroides,* especially *Alistipes*, consistent with several studies [58,59,61,62], but the change in *Firmicutes* was less consistent, as *Lactobacillus* decreased and *Subdilogranulum* increased. The genus *Alistipes* has been associated with the efficacy of checkpoint inhibitor immunotherapy in non-small cell lung cancer (NSCLC) and generally activation of innate immunity [65]. The large relative increase in *Verrucomicrobia*, particularly *Akkermansia muciniphila*, in obese mice, was surprising, as the abundance of *Akkermansia* has been shown to be inversely correlated with obesity in mice and humans [66]. *Akkermansia* is a mucolytic organism that has been associated with improved fasting glucose and reduced intestinal inflammation [66] and is decreased by high-fat diets [67]. It is possible that this relative increase in our model may be secondary to increased mucus production, which is stimulated by LPS and inflammatory cytokines [68]. Overnutrition is known to increase plasma LPS through altered intestinal barrier permeability [69]. A prolonged period of overnutrition with a Western diet may have produced sustained endotoxemia and a secondary increase in intestinal mucus. Notably, 16S sequence data show that in obese, tumor-bearing mice, the relative excess of *Akkermansia* disappeared and was replaced by *Firmicutes* such as *Clostridiaceae* and *Mogibacteriaceae*, producing a decreased *Bacteroides/Firmicutes* ratio that has been associated with obesity in other studies [58,59,60,61,62]. In our TNBC model, the presence of tumor had a far more dramatic effect on the intestinal microbiome in obese animals than in lean animals, even though no weight loss was associated with the presence of the tumor. The mechanism of this difference is unclear and may involve systemic metabolic changes and/or immunological changes related to the presence of the tumor. Very little is known about the interplay between TNBC and the intestinal microbiome. The possible role of the intestinal microbiome in the effectiveness of TNBC immunotherapy is being actively investigated [70,71], but no firm conclusions have been reached to date. Our results suggest that obesity may be an important modifier of the relationship between TNBC and the intestinal microbiome, which should be taken into consideration in clinical studies.

## 5. Conclusions

Obesity induced via “Western” diet in an immune-competent model of TNBC was associated enhanced tumor growth and with significant loss of diversity in the intestinal microbiome, and with a decrease in *Bacteroides* species, particularly *Alistipes.* Metabolic pathways in intestinal bacteria were also significantly affected by obesity, particularly in tumor-bearing animals. The contribution of the intestinal microbiome to tumor immunity and tumor growth in obese animals and patients with TNBC deserves further investigation.

## Figures and Tables

**Figure 1 nutrients-13-03656-f001:**
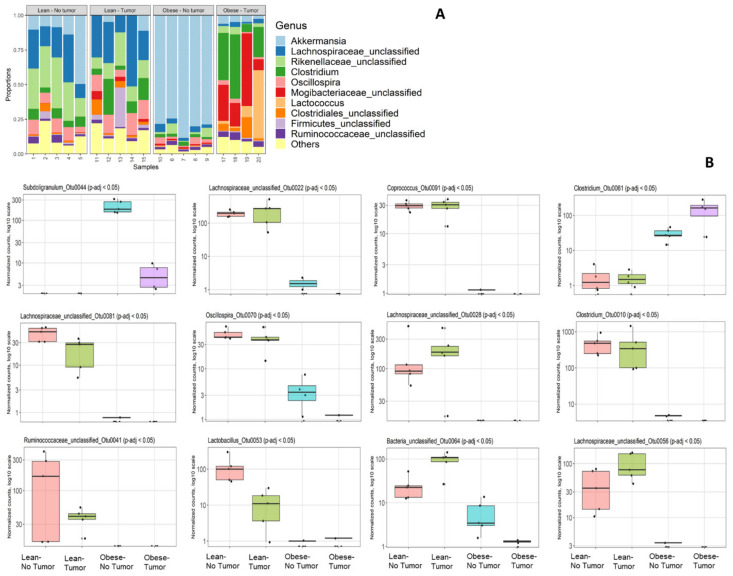
(**A**) Community composition by 16S rRNA sequencing. (**A**) Relative abundances of the most abundant taxa; (**B**) differential abundance testing: top 12 differentially abundant OTUs, sorted by q-value. In each plot, groups left to right are lean no tumor; lean–tumor; obese–no tumor; obese–tumor.

**Figure 2 nutrients-13-03656-f002:**
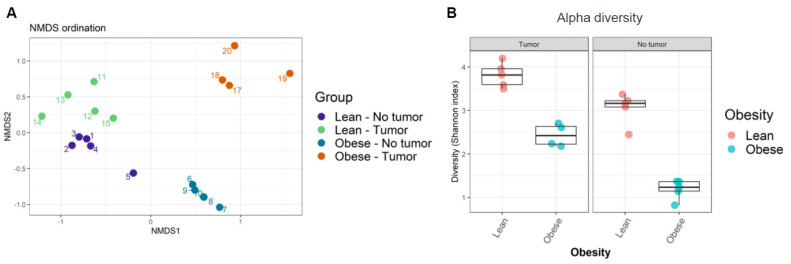
(**A**) Ordination plot displaying similarity of community composition between samples; (**B**) alpha diversity among samples was calculated using Shannon’s diversity index.

**Figure 3 nutrients-13-03656-f003:**
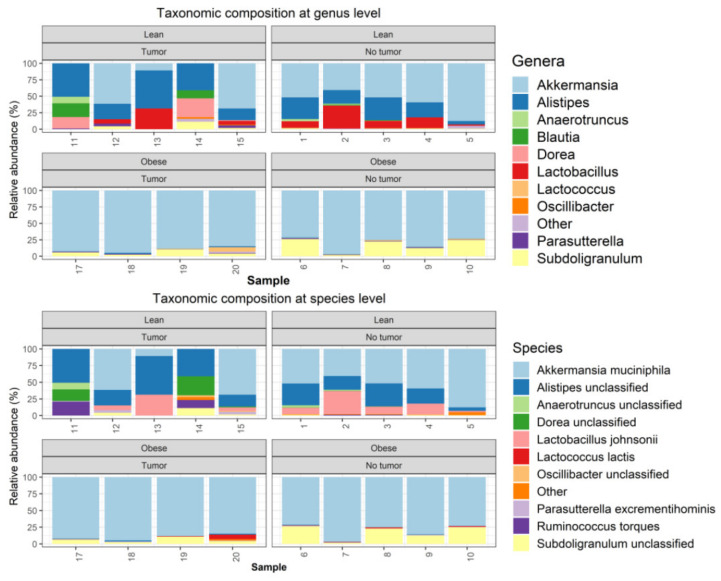
Metagenomics analysis of taxonomic composition at the genus and species level.

**Figure 4 nutrients-13-03656-f004:**
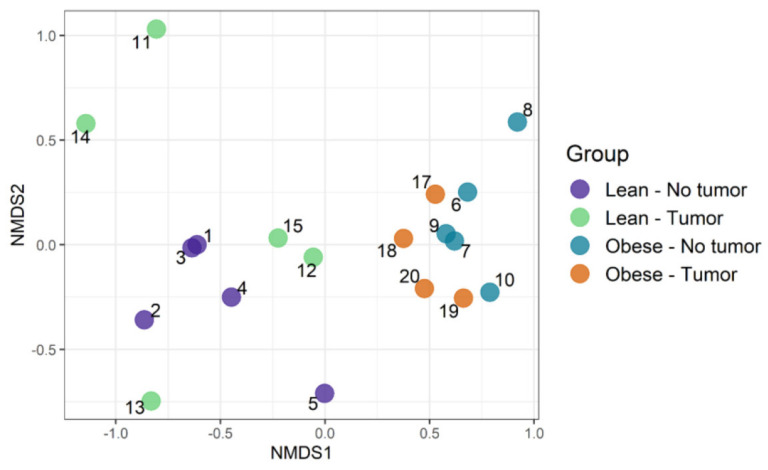
Community composition: visualizing similarity among microbiomes using an ordination plot.

**Figure 5 nutrients-13-03656-f005:**
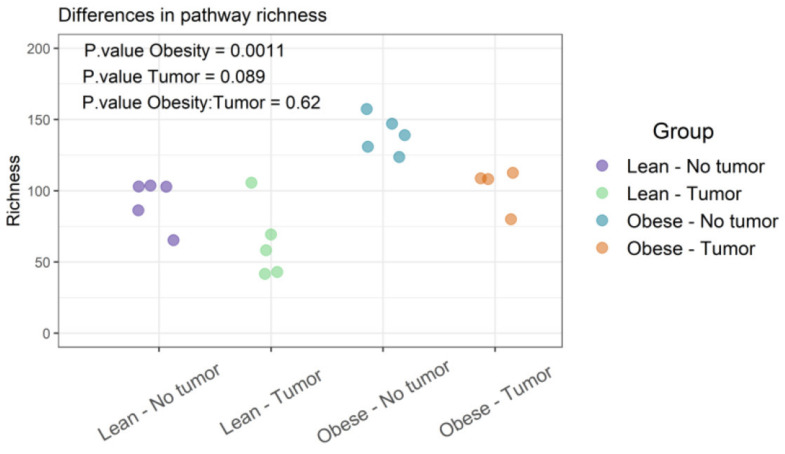
Functional profiles were summarized into pathways using the Metacyc pathway definition. Difference in pathway richness (number of unique pathways) was calculated using the Scheirer–Ray–Hare test and presented as richness plot.

**Figure 6 nutrients-13-03656-f006:**
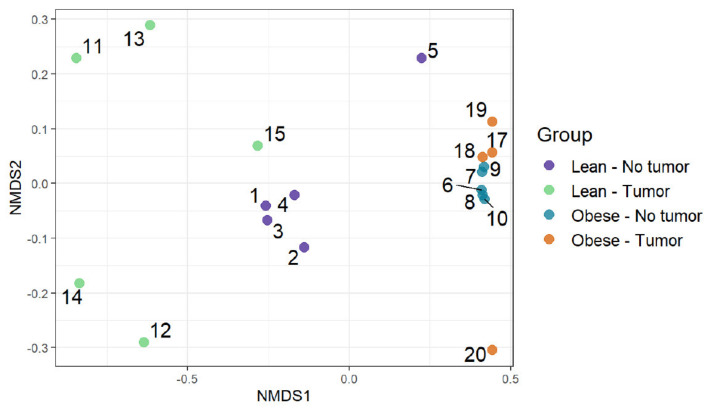
Functional profiles were summarized into pathways using the Metacyc pathway definition. Difference in pathway richness (number of unique pathways) was calculated using the Scheirer–Ray–Hare test and presented as a richness plot.

**Table 1 nutrients-13-03656-t001:** PERMANOVA (analysis of variance) of taxonomic groups: the significance of differences between treatments.

	Degrees of Freedom	Sum of Squares	F Model	R2	Pr (>F)
**Obesity**	1	1.133	16.831	0.457	0.0002
**Tumor**	1	0.156	2.321	0.063	0.1141
**Obesity:Tumor interaction**	1	0.180	2.672	0.073	0.0879
**Residuals**	15	1.010	NA	0.407	NA
**Total**	18	2.479	NA	1.000	NA

**Table 2 nutrients-13-03656-t002:** Summary of taxa that changed significantly according to obesity and tumor presence.

Taxa	*p*-Value Obesity	*p*-Value Tumor	*p*-Value Obesity–Tumor	Padj Obesity	Padj Tumor	Padj Obesity–Tumor
Alistipes unclassified	**0.0002**	0.9674	0.8396	**0.0028**	0.9674	0.8885
Ruminococcus torques	**0.0394**	0.8186	0.8288	0.0984	0.9096	0.8885
Dorea_unclassified	**0.0366**	0.9074	0.3451	0.0984	0.9552	0.5582
Eubacterium plexicaudatum	**0.0394**	0.3590	0.3872	0.0984	0.5733	0.5582
Lactobacillus johnsonii	**0.0011**	0.3627	0.3908	**0.0071**	0.5733	0.5582
Lactococcus_lactis	**0.0003**	0.4789	0.4527	**0.0028**	0.6842	0.6036
Oscillibacter unclassified	**0.0478**	0.6124	0.8885	0.1062	0.7403	0.8885
Subdoligranulum unclassified	**0.0141**	0.6039	0.0893	0.0565	0.7403	0.5582
Burkholderiales bacterium_1_1_47	**0.0030**	0.3057	0.2773	**0.0149**	0.5733	0.5582
Parasutterella excrementihominis	0.1318	0.2084	**0.0282**	0.2027	0.5733	0.5582
Akkermansia muciniphila	0.5675	**0.0239**	0.8850	0.5973	0.4778	0.8885

**Table 3 nutrients-13-03656-t003:** PERMANOVA: analysis of variation of functional groups.

	Degrees of Freedom	Sum of Squares	F Model	R2	Pr (>F)
Obesity	1	0.872	32.560	0.576	0.0001
Tumor	1	0.127	4.725	0.084	0.0196
Obesity–Tumor interaction	1	0.113	4.219	0.075	0.0295
Residuals	15	0.402	NA	0.265	NA
Total	18	1.513	NA	1.000	NA

**Table 4 nutrients-13-03656-t004:** Pathways with significantly different abundances.

Pathways with Significantly Different Abundances	*p*-Value Obesity	*p*-Value Tumor	*p*-Value Obesity Tumor
1CMET2-PWY: N10-formyl-tetrahydrofolate biosynthesis	**0.0010366**	0.635094	0.0717867
ANAEROFRUCAT-PWY: homolactic fermentation	**0.0009608**	0.4753113	0.4489497
ANAGLYCOLYSIS-PWY: glycolysis III (from glucose)	0.4510959	**0.0399166**	**0.0352809**
ARGSYN-PWY: L-arginine biosynthesis I (via L-ornithine)	**0.0001458**	0.8326309	0.7676368
ARGSYNBSUB-PWY: L-arginine biosynthesis II (acetyl cycle)	**0.0001458**	0.8326309	0.7676368
ARO-PWY: chorismate biosynthesis I	**0.0002371**	0.4868042	0.1897699
BRANCHED-CHAIN-AA-SYN-PWY: superpathway of branched amino acid biosynthesis	**0.0054811**	0.216359	0.1011976
COA-PWY-1: coenzyme A biosynthesis II (mammalian)	**0.0010853**	0.2266639	0.3961805
COA-PWY: coenzyme A biosynthesis I	**0.0002371**	0.4370334	0.2888817
COMPLETE-ARO-PWY: superpathway of aromatic amino acid biosynthesis	**0.0002326**	0.4364287	0.2201606
DENOVOPURINE2-PWY: superpathway of purine nucleotides de novo biosynthesis II	**0.0305188**	0.098992	0.7945958
DTDPRHAMSYN-PWY: dTDP-L-rhamnose biosynthesis I	**0.007936**	0.882567	0.7176804
FUCCAT-PWY: fucose degradation	0.1550304	0.2795522	**0.0181659**
GALACTUROCAT-PWY: D-galacturonate degradation I	**0.0197737**	0.4114757	0.540993
GLUCONEO-PWY: gluconeogenesis I	**0.0029886**	0.7411059	0.7260155
GLUCUROCAT-PWY: superpathway of &beta;-D-glucuronide and D-glucuronate degradation	0.1550304	0.2795522	**0.0181659**
GLUTORN-PWY: L-ornithine biosynthesis	**0.0001458**	0.8326309	0.7676368
GLYCOGENSYNTH-PWY: glycogen biosynthesis I (from ADP-D-Glucose)	**0.0477852**	0.0726354	0.2825296
GLYCOLYSIS: glycolysis I (from glucose 6-phosphate)	**0.0009608**	0.4753113	0.4489497
HISDEG-PWY: L-histidine degradation I	**0.0134637**	0.8473453	0.6476828

## Data Availability

Not applicable.

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
