# Peer review of "Obesity Modulates the Gut Microbiome in Triple-Negative Breast Cancer"

_nutrients, 2021, doi:10.3390/nu13103656_

Round 1

Reviewer 1 Report

Well done preclinical metagenomic profiling study which will have wide and meaningful interest in the oncology community. The methodology is clear, well presented to aide reproducibility. The publication of the Phase 3 Keynote-522 and 355 trials in TNBC.

I have a minor criticism of Table 2 where the column headers are too jumbled to be easily understood.

Good discussion with clear directionality for future research.

Author Response

We thank the Reviewer for this favorable assessment of our work. We have edited Table 2 to improve its layout and clarity

Reviewer 2 Report

This study aims to characterize the role of obesity in modulating the gut microbiome in a syngeneic mouse model of TNBC.

The paper  is well written and emphasizes the importance of obesity as an important risk factor in cancer. The authors analyze the taxonomic profile that varies significantly in the presence of the tumor

Moreover, the authors are familiar with the tumor model.

There are some things that need to be improved.

Figure 2B needs improvement is too small

Important assays are needed for factors linked to the condition of obesity:for example  insulin resistance, proinflammatory factors derived from adipose tissue, adiponectin and leptin.

Author Response

We thank the Reviewer for this favorable assessment of our work. We believe the reviewer was referring to Figure 1B, which we realize in the formatted version of the manuscript was far too small to read. We have replaced that figure with one where text is larger for ease of reading.

We completely agree with the reviewer that assays for adipokines including leptin would strengthen the manuscript. Unfortunately, the grants supporting this study do not support these tests. We have recently published on the effects of leptin from adipose stem cells on metastasis in TNBC models (Leptin produced by obesity-altered adipose stem cells promotes metastasis but not tumorigenesis of triple-negative breast cancer in orthotopic xenograft and patient-derived xenograft models - PubMed (nih.gov)